# Age Is Not So Important for Risk Stratification in Early Cholecystectomy for Acute Calculous Cholecystitis: A Post-Hoc Analysis of the SPRiMACC Study Database

**DOI:** 10.3390/medicina61071228

**Published:** 2025-07-07

**Authors:** Paola Fugazzola, Ahmed Ghaly, Luca Ansaloni, Francesca Dal Mas, Carlo Maria Bianchi, Enrico Cicuttin, Andrea Dagnoni, Simone Frassini, Matteo Tomasoni, Lorenzo Cobianchi

**Affiliations:** 1General Surgery Department, Fondazione IRCCS Policlinico San Matteo, 27100 Pavia, Italy; paola.fugazzola@gmail.com (P.F.); l.ansaloni@smatteo.pv.it (L.A.); cm.bianchi@smatteo.pv.it (C.M.B.); enrico.cqtn@gmail.com (E.C.); andrea.dagnoni01@universitadipavia.it (A.D.); simone.frassini01@universitadipavia.it (S.F.); matteotomasoni83@gmail.com (M.T.); lorenzo.cobianchi@unipv.it (L.C.); 2Department of Clinical, Surgical, Diagnostic & Pediatric Sciences, University of Pavia, 27100 Pavia, Italy; 3Venice School of Management, Ca’ Foscari University of Venice, 30123 Venice, Italy; francesca.dalmas@unive.it; 4Collegium Medicum, University of Social Sciences, 90-237 Lodz, Poland

**Keywords:** early cholecystectomy (EC), acute calculous cholecystitis (ACC), world society of emergency surgery (WSES), Tokyo guidelines 18 (TG18)

## Abstract

*Background and Objectives:* Early cholecystectomy (EC) is widely regarded as the first-line treatment for acute calculous cholecystitis (ACC). Current debate centers on the feasibility of EC as an option even for elderly patients. This study aims to determine whether age alone is an independent risk prediction factor for prognosis after EC for ACC. *Materials and Methods:* This study is a post-hoc analysis of the S.P.Ri.M.A.C.C. WSES prospective international multicenter observational study database, including patients with ACC undergoing EC. Univariate and multivariate analyses were conducted, examining different risk factors for major morbidity and mortality after EC. *Results:* In the univariate analyses, age was found to be a statistically significant risk factor for both 30-day major complications (*p* < 0.001) and 30-day mortality (*p* = 0.003). However, in the multivariate analysis, age alone was not a significant predictor for either outcome, with *p*-values of 0.419 and 0.094, respectively. The only significant risk factor associated with both 30-day mortality and major morbidity in the multivariate model was the POSSUM Physiological Score (PS). *Conclusions:* Age alone cannot be considered a reliable risk predictor for a complicated postoperative course after EC in patients with ACC. Frailty, rather than chronological age, should be assessed to predict the outcome of these patients.

## 1. Introduction

The prevalence of gallstones in the general population is estimated to be 10–15%, with approximately 20–40% of these individuals developing gallstone-related complications [1]. Acute calculous cholecystitis (ACC) represents the initial complication in 10–15% of such cases [2]. The Tokyo guidelines 2018 (TG18) [3,4,5] and the World Society of Emergency Surgery 2020 guidelines (WSES GL) [1,6] are the leading protocols for managing ACC, with both recommending early cholecystectomy (EC) as the primary treatment for ACC. However, considerable debate persists over the selection criteria for high-risk patients undergoing EC, particularly concerning whether age should be a limiting factor [7].

Several established risk prediction models—including the POSSUM [8,9], the modified Frailty Index (mFI) [10], the Charlson Comorbidity Index (CCI) [11], and the Chole-Risk score [12]—have been applied and validated to identify high-risk patients for adverse outcomes after EC in ACC. Nonetheless, most of these models lack formal prospective or external validation [7].

Recently, the Scores for Prediction of Risk for postoperative major Morbidity after cholecystectomy in Acute Calculous Cholecystitis (SPRiMACC) study identified the POSSUM Physiological Score (PS) as the most effective predictor of complicated outcomes after EC in ACC patients [7]. The study proposed a PS cutoff of 25 to identify high-risk patients [7]. However, some researchers consider the use of these scores in an emergency setting to be overly time-consuming and complex. This has led to efforts to simplify risk assessments, with an increasing focus on age as a primary factor [13].

A recent study by Matsui et al. [13] proposed that risk stratification for EC in ACC patients could be simplified by focusing on age, ASA-PS, and TG18 ACC grade. The study concluded that detailed risk assessment should primarily target elderly patients.

On the other hand, the study by Di Martino et al. [14] indicated that while EC for ACC in patients aged 85 and older is associated with significant morbidity and mortality, it remains a safe option for selected patients. Therefore, age alone should not be considered a reason to avoid EC for ACC.

A systematic review by Montenegro et al. [15] supports laparoscopic cholecystectomy as a generally safe approach for elderly patients, advising against conservative treatment. Furthermore, many studies in the literature show how delaying surgery after ACC in the elderly population is an unsuccessful strategy, as many patients do not undergo surgery by the end of the follow-up period. According to the study by Giraud et al. [16], elective cholecystectomy after ACC failed in 44% of the patients aged > 75 years, for various reasons. This failure in performing elective surgery not only complicates a non-ideal situation, but it also exposes patients to the risk of recurrence. Indeed, the risk of developing symptoms again among those who were not operated on is 39% over 2 years [17]. Additionally, the conservative management of older patients with acute cholecystitis relates to a lower probability of spontaneous resolution of symptoms compared to younger patients [17,18].

The CHOCOLATE trial [19] demonstrated that even for high-risk patients with acute cholecystitis, laparoscopic cholecystectomy is preferred over percutaneous drainage from both clinical and economic perspectives.

Many clinicians continue to use chronological age as a major predictor for surgical risk, especially in emergency procedures like early cholecystectomy for acute calculous cholecystitis. However, this approach is increasingly questioned. Studies suggest that physiological status and comorbidity burden, rather than age alone, play a more significant role in determining postoperative outcomes [6,7,19]. In particular, the integration of objective scoring systems—such as the Charlson Comorbidity Index (CCI), ASA physical status classification, and POSSUM Physiological Score (PS)—has improved our ability to stratify patients more accurately [20]. These tools help capture the complexity of surgical risk in a way that age alone cannot, especially in the setting of acute care surgery where timely intervention is critical [19].

Given the ongoing debate, this study aims to assess whether physiological and frailty-based scores provide superior predictive value over chronological age in stratifying surgical risk for high-risk ACC patients undergoing early cholecystectomy. To this end, we conducted a post-hoc analysis of the SPRIMACC database [7].

## 2. Materials and Methods

### 2.1. Ethical Considerations

The present post hoc analysis received ethical approval from the medical ethics board of the coordinating centre, the IRCCS San Matteo University Hospital in Pavia, Italy. Prior to enrollment in the original SPRiMACC study, from which this analysis derives, all patients provided both verbal and written informed consent. The study was conducted in accordance with the principles of the Declaration of Helsinki.

### 2.2. Design

This study represents a post-hoc analysis of the SPRiMACC study dataset, aimed at evaluating the role of age as a predictor of high postoperative risk in ACC patients undergoing EC, and assessing whether age alone could serve as a reliable criterion for excluding patients from emergency cholecystectomy. The SPRiMACC study is a multicenter, observational, prospective study promoted by the World Society of Emergency Surgery (WSES). Conducted from 1 September 2021 to 1 September 2022, a total of 1253 patients were enrolled from 79 centers across 19 countries. The SPRiMACC study is registered on LegalTrial.gov under case number NCT04995380. Patient recruitment took place in the preoperative phase, conducted by surgeons at participating centers. Enrollment was based on the patients’ clinical conditions and the results of instrumental and biochemical investigations, confirming the ACC diagnosis. The full SPRiMACC study protocol is available on the study website https://sprimaccstudy.wixsite.com/website (accessed on 11 October 2023) [7].

The primary aim of the SPRiMACC study was to prospectively validate and compare the effectiveness of preoperative risk prediction models—including the Chole-Risk score, POSSUM Physiological Score (PS), modified Frailty Index (mFI), Charlson Comorbidity Index (CCI), American Society of Anesthesiologists-Performance Status (ASA-PS), APACHE II score, and ACC severity grade—in predicting in-hospital mortality, 30-day mortality, in-hospital major morbidity (Clavien-Dindo ≥ 3), and 30-day major morbidity in patients with ACC undergoing EC [7].

### 2.3. Studied Variables

In this analysis, eight variables were evaluated: age, BMI, POSSUM-PS, PCT, sex, TG18 ACC grade, time from symptom onset to EC, and the operative duration. Additional variables related to comorbidities and the clinical status were available in the SPRiMACC database. However, given that the POSSUM-PS accounts for these factors and is recognized as a reliable prognostic model, it was selected as the sole indicator of frailty, summarizing all related variables [20].

### 2.4. Outcomes

The primary outcomes analyzed were 30-day major complications and 30-day mortality rate. Major complications were considered for those with a Clavien Dindo grade > 2 (3, 4, and 5).

### 2.5. Inclusion and Exclusion Criteria

Inclusion criteria of the SPRiMACC study were as follows:

(1)diagnosis of ACC as defined by the 2018 TGs criteria,(2)candidacy for EC during the index admission (any surgical technique, laparoscopic or open procedures, including also bailout procedures such as subtotal cholecystectomy),(3)age ≥ 18 years old,(4)risk stratification for associated common bile duct stones, and, in case of confirmation, preoperative ERCP,(5)provision of a signed and dated informed consent form,(6)willingness to comply with all study procedures and be available for the duration of the study [7].

Exclusion criteria were:(1)pregnancy or lactation,(2)ACC is not related to gallstone etiology,(3)symptoms onset > 10 days before cholecystectomy (patients with ACC associated with common bile duct stones were included if they received EC within 10 days of symptom onset following preoperative ERCP),(4)concomitant cholangitis or pancreatitis,(5)intraoperative treatment of common bile duct stones,(6)any condition that would increase the risk for the patient or preclude the individual’s full compliance or completion of the study [7].

### 2.6. Statistical Analysis

A univariate analysis was conducted for each considered variable relative to both outcomes. Univariate comparisons were performed using Student’s *t*-test for continuous variables with normal distribution and the Mann–Whitney test for non-normal distribution variables. For testing for normality, the Shapiro–Wilk test was used. Categorical variables were compared using the chi-square or Fisher’s exact test, as appropriate. Significant variables identified in the univariate analysis were subsequently analyzed using binomial logistic regression to identify independent risk factors for the studied outcomes. Statistical significance was defined as *p* ≤ 0.05.

## 3. Results

Our study, utilizing data from the SRiMACC database, included 1253 patients from 79 centers across 19 different countries, enrolled from 1 September 2021 to 1 September 2022.

Among the 1253 patients included in the SPRiMACC cohort, the mean age was 59.4 ± 17.0 years (range 18–98), with a slight predominance of intermediate clinical severity according to the TG18 ACC grading system: 67.3% were classified as Grade 2, 32.2% as Grade 1, and only 0.5% as Grade 3. Regarding preoperative anesthetic risk, ASA scores were predominantly 2 (42.2%) and 3 (25.5%), while 21.1% were ASA 1. The Charlson Comorbidity Index (CCI) had a mean of 2.5 ± 2.4, and the modified Frailty Index (m-FI) was 1.0 ± 1.4. The POSSUM Physiological Score (PS), central to our multivariate analysis, had a mean value of 20.8 ± 6.5 (range 12–59). Additional scoring systems reported included Chole-Risk (with most patients in score categories 0–2) and APACHE II, with a mean of 6.6 ± 5.1.

The patients’ preoperative characteristics and the scores are shown in Table 1.

The 30-day major complication (CD > 2) rate and the 30-day mortality rate were 6.6% and 1.1%, respectively.

Data for the eight variables were analyzed and reported in Table 2 and Table 3 for 30-day major complications (CD > 2) and 30-day mortality, respectively.

The univariate analysis identified age as a significant risk factor for both 30-day major morbidity (OR 1.029, 95% CI 1.014–1.044, *p* < 0.001) and 30-day mortality (OR 1.066, 95% CI 1.021–1.112, *p* = 0.003). Other significant risk factors for major morbidity in univariate analysis were the PCT value (OR 0.017, 95% CI 1.002–1.016, *p* = 0.017), the TG 18 ACC grade (OR 2.340, 95% CI 1.338–4.090, *p* = 0.003), the POSSUM PS (OR 1.097, 95% CI 1.065–1.130, *p* < 0.001) and the operative time (OR 1.010, 95% CI 1.006–1.014, *p* < 0.001).

Significant risk factors for 30-day mortality included age (OR 1.066, 95% CI 1.021–1.112, *p* = 0.003), POSSUM-PS (OR 1.218, 95% CI 1.142–1.298, *p* < 0.001), PCT (OR 1.013, 95% CI 1.004–1.022, *p* = 0.007) and TG 18 ACC grade (OR 79.492, 95% CI 15.160–416.810, *p* < 0.001).

However, the only significant risk factor for 30-day mortality at multivariate analysis was the POSSUM-PS. POSSUM-PS showed an OR of 1.131 (95% CI 1.034–1.237, *p* = 0.007) for 30-day mortality. Regarding the 30-day major complications outcome, POSSUM-PS and operative time were the only significant risk factors at multivariate analysis. The POSSUM-PS showed an OR of 1.083 (95% CI 1.038–1.130, *p* < 0.001).

Interestingly, several variables that showed statistically significant associations with adverse outcomes in the univariate analyses did not maintain their predictive value in the multivariate models. For instance, age was significantly associated with both 30-day major complications (OR 1.029, 95% CI 1.014–1.044; *p* < 0.001) and 30-day mortality (OR 1.066, 95% CI 1.021–1.112; *p* = 0.003) in univariate analysis. However, it did not reach significance in the adjusted models (*p* = 0.419 for complications; *p* = 0.094 for mortality), suggesting that age’s effect is likely mediated through other clinical variables. Similarly, procalcitonin (PCT) was associated with both endpoints in univariate testing (*p* = 0.017 and *p* = 0.007), but failed to remain significant in multivariate models (*p* = 0.206 and *p* = 0.333, respectively). Additionally, TG18 ACC grade, while significant for both major complications (*p* = 0.003) and mortality (*p* < 0.001) in univariate analysis, did not retain significance in multivariate models (*p* = 0.355 and *p* = 0.118, respectively). These findings underscore the importance of adjusting for confounders when assessing prognostic factors and reinforce the value of comprehensive scores like the POSSUM-PS, which integrate multiple dimensions of patient risk.

## 4. Discussion

Elective laparoscopic cholecystectomy is now widely considered a safe and feasible procedure in the elderly population, with age no longer considered a contraindication due to advances in surgical and anesthesiology techniques that have significantly improved operative outcomes [4,18,21]. However, in the context of acute calculous cholecystitis (ACC), in an emergency setting, some authors continue to regard age as a negative prognostic factor for outcomes following early cholecystectomy, despite substantial evidence suggesting that the focus should shift from age to frailty [14,18,22,23]. A recent study by Matsui et al. identified age and TG18 ACC grade as independent risk factors within their cohort, and, when splitting the cohort into age groups, they showed statistical significance for the elderly group (patients aged 65 to 79 years old) [13]. They conclude that the risk stratification for EC in ACC patients could be streamlined by focusing on age, ASA-PS, and TG18 ACC grade: the risk increases as the patient’s age rises and ASA-PS deteriorates, particularly when the severity of the TG18 ACC grade is high. Therefore, the detailed risk assessment should focus on the elderly population. The Matsui et al. study is a retrospective single-centre study with a relatively small sample size of 350 patients.

On the other hand, our post-hoc analysis of the SPRiMACC database—which includes a broad cohort of patients from multiple international centers—did not confirm these findings. According to the results of our study, neither age nor TG18 ACC grade can be considered an independent risk factor; the only independent significant risk factor for both morbidity and mortality is the CHOLE-POSSUM (POSSUM-PS with a threshold score ≥ 25). Age is a component of the CHOLE-POSSUM; the strength of this score lies in being specifically tailored for acute calculous cholecystitis (ACC), considering not only comorbidities (which age indexes) but also current inflammatory and infectious conditions.

There is a notable distinction between chronological age and frailty, particularly in urgent medical contexts where frailty is influenced not only by comorbidities but also by the patient’s acute inflammatory and infectious conditions [24,25]. Although frailty is often associated with advancing age, it is not exclusive to the elderly, nor is every older patient frail. Studies have demonstrated diverse rates of frailty across different surgical disciplines, highlighting its diverse prevalence in clinical contexts [26].

Furthermore, our analysis highlights that some traditionally considered risk factors—such as age, TG18 grade, and procalcitonin (PCT) levels—were significantly associated with adverse outcomes only in univariate analysis. The loss of statistical significance in multivariate models suggests that these variables are not independent predictors of complications or mortality but rather act as proxies for more complex clinical conditions already captured by composite scores like the POSSUM Physiological Score. This finding reinforces the value of multidimensional tools in surgical risk stratification.

A closer look at the preoperative characteristics of our cohort (Table 1) reveals a wide variability in physiological status, with POSSUM-PS values ranging from 12 to 59 and a mean of 20.8 ± 6.5. This heterogeneity is further supported by the distribution of ASA and CCI scores, and by the average modified Frailty Index (m-FI) of 1.0 ± 1.4. These data strengthen the notion that chronological age alone lacks the granularity to reflect a patient’s true operative risk, especially in the context of acute care surgery.

Finally, our study shows that among all variables assessed, only the POSSUM-PS and operative time remained independent predictors of major complications in the multivariate model. This underscores the importance of thoroughly evaluating physiological reserve before proceeding with surgery, particularly in high-risk patients. Early identification of vulnerable individuals using objective tools like the POSSUM-PS may support better perioperative planning and more nuanced, shared decision-making among surgeons, anesthesiologists, and patients.

Recent evidence further supports the notion that chronological age alone should not dictate surgical decision-making. Giraud et al. conducted a retrospective study on patients over 75 years of age with acute cholecystitis and found that while elective cholecystectomy strategies often failed in the elderly, the failures were primarily related to frailty, comorbidities, or personal refusal rather than age per se. Their findings reinforce that advanced age, in isolation, is not a predictor of poor surgical outcomes, and emphasize the necessity of individualized risk assessments based on physiological reserve and patient wishes rather than age alone [16].

Our findings are further supported by recent case-based evidence showing that age, in isolation, may obscure critical risk factors such as chronic inflammation or comorbidities like diabetes mellitus. Zanchetta et al. described a 71-year-old patient with previously asymptomatic cholelithiasis who presented with perforated ACC and an incidental squamous cell carcinoma of the gallbladder, emphasizing the risks of delaying surgical intervention in elderly patients [27].

The limit of our study is its nature as a post-hoc analysis, essentially a retrospective re-analysis of an existing dataset collected prospectively to assess operative risk factors, including age. However, the large sample size and the difficulties in performing randomized controlled studies in the emergency setting make this study the highest level of available evidence on this specific topic.

## 5. Conclusions

Chronological age alone is not an adequate predictor of risk in patients undergoing early cholecystectomy for acute calculous cholecystitis. Surgical decision-making should instead be guided by validated risk models like the POSSUM-PS, which better capture the complexity of patient condition in the acute setting. Our findings advocate for a more individualized and evidence-based approach, whereby age is considered within the broader context of frailty and physiological reserve rather than used as a binary exclusion criterion.

## 6. Future Directions

While early cholecystectomy remains the standard treatment for acute calculous cholecystitis (ACC), evolving minimally invasive alternatives may provide significant benefit, especially for high-risk patients. Among these, EUS-guided gallbladder drainage (EUS-GBD) with Lumen-Apposing Metal Stents (LAMS) is emerging as a promising approach [28].

To further investigate this approach, our group is currently conducting a randomized controlled trial comparing EUS-GBD with LAMS to early cholecystectomy in high-risk patients presenting with acute cholecystitis, with a focus on safety, efficacy, and clinically relevant outcomes in real-world practice.

## Figures and Tables

**Table 1 medicina-61-01228-t001:** Preoperative characteristics of patients.

SCORE	% (N = 1253) Mean ± SD (Min-Max)
TG18 ACC grade	
− 1	32.2
− 2	67.3
− 3	0.5
ASA	
− 1	21.1
− 2	42.2
− 3	25.5
− 4	3.8
− 5	0.2
Chole-Risk	
− 0	18.0
− 1	44.4
− 2	25.9
− 3	7.4
− 4	0.8
m-FI	1.0 ± 1.4 (0–8)
CCI	2.5 ± 2.4 (0–19)
POSSUM PS	20.8 ± 6.5 (12–59)
APACHE II	6.6 ± 5.1 (0–71)
Age	59.4 ± 17.0 (18–98)

**Table 2 medicina-61-01228-t002:** Univariate and multivariate analysis of 30-day major complications (CD > 2).

	Univariate		Multivariate	
Variable	OR (95% CI)	*p*-Value	OR (95% CI)	*p*-Value
Age	1.029 (1.014–1.044)	<0.001	1.009 (0.987–1.032)	0.419
BMI	1.010 (0.963–1.060)	0.678	-	-
POSSUM PS	1.097 (1.065–1.130)	<0.001	**1.083 (1.038–1.130)**	**<0.001**
PCT	1.009 (1.002–1.016)	0.017	1.005 (0.997–1.014)	0.206
Sex	0.631 (0.396–1.003)	0.051	-	-
TG 18 ACC grade	2.340 (1.338–4.090)	0.003	1.433 (0.668–3.071)	0.355
Time from onset of symptoms to EC (days)	1.000 (0.994–1.006)	0.971	-	-
Operative time	1.010 (1.006–1.014)	<0.001	1.011 (1.005–1.017)	<0.001

**Table 3 medicina-61-01228-t003:** Univariate and multivariate analysis of 30-day mortality.

	Univariate		Multivariate	
Variable	OR (95% CI)	*p*-Value	OR (95% CI)	*p*-Value
Age	1.066 (1.021–1.112)	0.003	1.069 (0.989–1.155)	0.094
BMI	0.906 (0.778–1.055)	0.205	-	-
POSSUM-PS	1.218 (1.142–1.298)	<0.001	**1.131 (1.034–1.237)**	**0.007**
PCT	1.013 (1.004–1.022)	0.007	1.007 (0.993–1.021)	0.333
Sex	0.304 (0.084–1.095)	0.069	-	-
TG 18 ACC grade	79.492 (15.160–416.810)	<0.001	5.254 (0.655–42.144)	0.118
Time from onset of symptoms to EC (days)	0.979 (0.994–1.006)	0.713	-	-
Operative time	1.004 (0.995–1.014)	0.378	-	-

## Data Availability

The original contributions presented in this study are included in the article and in the SPRiMACC study. Further inquiries can be directed to the corresponding author.

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
