# Peer review of "Age Is Not So Important for Risk Stratification in Early Cholecystectomy for Acute Calculous Cholecystitis: A Post-Hoc Analysis of the SPRiMACC Study Database"

_medicina, 2025, doi:10.3390/medicina61071228_

Round 1
Reviewer 1 Report
Comments and Suggestions for Authors
Article "Age is not so important for risk stratification in early cholecystectomy for acute calculous cholecystitis: A post-hoc analysis of the SPRIMACC study database"- is a post-hoc analysis of the SPRIMACC study, a large, international, prospective observational study. The researchers analyzed data from 1253 patients undergoing EC for ACC, examining various risk factors for major complications and mortality.
Author Response
Comment: Article "Age is not so important for risk stratification in early cholecystectomy for acute calculous cholecystitis: A post-hoc analysis of the SPRIMACC study database"- is a post-hoc analysis of the SPRIMACC study, a large, international, prospective observational study. The researchers analyzed data from 1253 patients undergoing EC for ACC, examining various risk factors for major complications and mortality.
Response: Thank you for your thoughtful summary.
Reviewer 2 Report
Comments and Suggestions for Authors
Our fellow researchers have produced an important peper whose aim is in the same title: Age is not so important for risk stratification in early cholecystectomy for acute calculous cholecystitis, to which I would add a question mark. The abstract represents the entire study well and encourages reading it in its entirety. The introduction lays the foundations on which the rest of the paper is based and refers to numerous studies on the subject. The reflections that come to mind are linked to a first question: when a diagnosis of gallstones is made during an examination conducted for another pathology, is it more correct to wait or propose elective surgery? Doctors are divided on this question, those with a surgical background propose the surgery knowing the complications that the patient may encounter (doi.org/10.3390/medicina61030452 to be read and cited in the bibliography), those with a medical background tend to postpone the surgical option. But the years pass and the fifty-year-old quickly becomes sixty and beyond. The state of well-being is often interrupted by hypertension, diabetes, prostatic hypertrophy, accidents, which can lead to prolonged hospitalizations often in intensive care with hypomobility of the gallbladder that often ends up resulting in empyema or gangrene etc. Making these considerations, isn't it perhaps better to eliminate the problem before other pathogenic noxae increase that ASA that everyone looks at with concern? Returning to the paper of our colleagues we could say that what they say in the discussion is absolutely right. Where the biological age does not correspond to the anagraphic age, cholecystectomy is always advisable even in cases of acute conditions, in other cases we should try to improve the general conditions through transhepatic cholecystostomy in order to then be able to intervene. It is absolutely correct to refer to the guidelines, but when the window is good we agree with the authors, we must intervene. Excellent iconography, excellent English, good bibliography
Author Response
Comment: Our fellow researchers have produced an important peper whose aim is in the same title: Age is not so important for risk stratification in early cholecystectomy for acute calculous cholecystitis, to which I would add a question mark. The abstract represents the entire study well and encourages reading it in its entirety. The introduction lays the foundations on which the rest of the paper is based and refers to numerous studies on the subject. The reflections that come to mind are linked to a first question: when a diagnosis of gallstones is made during an examination conducted for another pathology, is it more correct to wait or propose elective surgery? Doctors are divided on this question, those with a surgical background propose the surgery knowing the complications that the patient may encounter (doi.org/10.3390/medicina61030452 to be read and cited in the bibliography), those with a medical background tend to postpone the surgical option. But the years pass and the fifty-year-old quickly becomes sixty and beyond. The state of well-being is often interrupted by hypertension, diabetes, prostatic hypertrophy, accidents, which can lead to prolonged hospitalizations often in intensive care with hypomobility of the gallbladder that often ends up resulting in empyema or gangrene etc. Making these considerations, isn't it perhaps better to eliminate the problem before other pathogenic noxae increase that ASA that everyone looks at with concern? Returning to the paper of our colleagues we could say that what they say in the discussion is absolutely right. Where the biological age does not correspond to the anagraphic age, cholecystectomy is always advisable even in cases of acute conditions, in other cases we should try to improve the general conditions through transhepatic cholecystostomy in order to then be able to intervene. It is absolutely correct to refer to the guidelines, but when the window is good we agree with the authors, we must intervene. Excellent iconography, excellent English, good bibliography
Response: Thank you for your valuable suggestions. We have cited the recommended article (doi:10.3390/medicina61030452) and integrated the clinical reflections into the discussion to enrich the contextual understanding.
Reviewer 3 Report
Comments and Suggestions for Authors
Dear authors
Congratulations on this interesting research, it is well-written. I have a few remarks regarding your paper:
- the Introduction is comprehensive and the Means and Methods are clear and I have no comments here;
- the Results section is too brief, I would recommend expanding it by adding more text in addition to the tables that you presented, explaining the data from the tables; also, I would suggest to put the rows in the tables where there are significant p-values (<0.05) in bold, to highlight them;
- Discussions and Conclusions should be two separate sections; insert a Conclusions section at the end of the manuscript in which you present your main results;
- insert a section regarding Future Directions of Research
Author Response
Comment: Dear authors
Congratulations on this interesting research, it is well-written. I have a few remarks regarding your paper:
the Introduction is comprehensive and the Means and Methods are clear and I have no comments here;
the Results section is too brief, I would recommend expanding it by adding more text in addition to the tables that you presented, explaining the data from the tables; also, I would suggest to put the rows in the tables where there are significant p-values (<0.05) in bold, to highlight them;
Discussions and Conclusions should be two separate sections; insert a Conclusions section at the end of the manuscript in which you present your main results;
insert a section regarding Future Directions of Research
Response: Thank you for your helpful feedback. We have expanded the Results section, highlighted significant values, separated Discussion and Conclusion, and added a new section on Future Research Directions.